# Effect of Droplet Size Parameters on Droplet Deposition and Drift of Aerial Spraying by Using Plant Protection UAV

**Shengde Chen** [1,2], **Yubin Lan** [1,2,*], **Zhiyan Zhou** [2,3], **Fan Ouyang** [1,2], **Guobin Wang** [4], **Xiaoyu Huang** [2,3], **Xiaoling Deng** [1,2] **and Shengnan Cheng** [2,3]

[1] College of Electronic Engineering, South China Agricultural University, Guangzhou 510642, China; shengde-chen@scau.edu.cn (S.C.); ouyangfan@scau.edu.cn (F.O.); dengxl@scau.edu.cn (X.D.)

[2] National Center for International Collaboration Research on Precision Agricultural Aviation Pesticides Spraying Technology (NPAAC), Guangzhou 510642, China; zyzhou@scau.edu.cn (Z.Z.); hsmy_huang@163.com (X.H.); 15915167024@163.com (S.C.)

[3] College of Engineering, South China Agricultural University, Guangzhou 510642, China

[4] School of agricultural Engineering and food science, Shandong University of Technology, Zibo 255022, China; guobinwang@sdut.edu.cn

[*] Correspondence: ylan@scau.edu.cn; Tel.: +86-020-8528-1421

**Abstract:** In the field of pesticide spraying, droplet size is one of the most important factors affecting droplet deposition and drift. In order to study the effect of different droplet size parameters on droplet deposition distribution and drift of aerial spraying by using plant protection UAV, an aerial spraying test with the same spraying rate and different size droplets in rice canopy was carried out by using multi-rotor unmanned aerial vehicles (UAV) and four TEEJET nozzles with different orifice sizes (these droplets with a volume median diameter (VMD) of 95.21, 121.43, 147.28, and 185.09 μm, respectively), and the deposition distribution and penetration of droplets in the target area and the drift distribution of droplets in the non-target area were compared and analyzed. The results showed that the deposition distribution and penetration of droplets in the target area and the drift distribution of droplets in the non-target area were influenced by the droplet size. The droplet deposition rate in the upper and lower rice canopies were increased in the target area with the increase of droplet size. The penetration results of droplets also increased with the increase of droplet size, and that of droplets with a VMD of 185.09 μm was the best, reaching 38.13%. The average values of the cumulative drift rate of droplets in the rice canopy in the four tests were 73.87%, 50.26%, 35.91%, and 23.06%, respectively, and the cumulative drift rate and the drift distance of droplets decreased with the increase of droplet size, which indicated that the increase of droplet size can effectively reduce droplet drift. It demonstrated that the droplet size is one of the most important factors affecting droplet deposition and drift for pesticide spraying by plant protection UAV, and for the application of plant protection UAV with extra-low volume spraying, the use of droplets with VMD less than 160 μm should be avoided and a more than 10 m buffer zone should be considered downwind of the spraying field to avoid drug damage caused by pesticide drift. The results have fully revealed the effect of droplet size parameters on droplet deposition and drift of aerial spraying. Moreover, the influence of the wind field below the rotors on the distribution of droplet deposition was surmised and analyzed from the perspective of plant protection UAV. It is important for optimizing the droplet parameters of aerial spraying, increasing the spraying efficiency, and realizing precision agricultural aviation spray.

**Keywords:** plant protection UAV; pesticide spraying; droplet deposition; pesticide drift; spraying efficiency

## 1. Introduction

Chemical control for diseases, insects, and weeds is an important agricultural production technology, and it is estimated that agrochemicals prevent up to 45% of loss from of the world's food supply [1,2]. In the current production process of crops, the traditional manual and semi-mechanical operations continue to be the main methods of plant protection in China, which not only leads to the excessive spray and low utilization rate of pesticides, but also causes a large amount of pesticide residues [3,4]. Even worse, the excessive spraying of pesticides can seriously pollute the ecological environment and threaten food and human safety. According to statistics, the pesticide use per unit area in China is 2.5 times higher than the world average, and the contaminated cultivated land area has reached $1 \times 10^7$ hm$^2$, accounting for about 1/10 of the arable area (Data from the Ministry of Ecology and Environment of the People's Republic of China).

As a new type of plant protection in China in recent years, the low-altitude and low-volume spraying technology by using plant protection unmanned aerial vehicles (UAV) has advantages over traditional spraying methods since it is efficient and allows for a quick response to sudden pest outbreaks [5]. Labor costs for operation are also low, and crops and the physical structure of soil are not damaged by equipment [6]. Furthermore, aerial spraying can reduce pesticide application by 15–20% by using a low or ultra-low volume of spray, and it can be used as an important technical support for a reduction program for chemical fertilizers and pesticides [7,8]. Thus, the aerial spraying operation by using plant protection UAV has gradually become the preferred method for plant protection operations in China [9].

With the rapid development and application of plant protection UAV in China, low-altitude and low-volume spraying technology has become a research hotspot. A series of explorations have been made by researchers on the quality of its operation and the effect of droplet deposition distribution. Qiu et al. [10] studied the relationship between spray deposition characteristics and flight height and velocity of a small unmanned helicopter and developed a model of the relationship between droplet deposition distribution and the two factors of flight height and velocity. Xue et al. [11] performed a series of field trials about UAV spraying to evaluate various techniques (flew at a height of 5 m, a speed of 3 m/s, and wind speed of 3 m/s) for measuring droplet deposition and drift during spray application to a paddy field. Chen et al. [12] studied the effect of different spray operation parameters on the droplet deposition distribution of aerial spraying in the hybrid rice canopy by the single-rotor unmanned helicopter. Qin et al. [13] demonstrated the influence of spray parameters (such as droplets of 480 g/L chlorpyrifos (Regent EC), flight height of 0.8 m and 1.5 m, flight velocity of 3 m/s and 5 m/s) of the UAV on spray deposition in a rice canopy for control of plant hoppers, showing that low-volume applications of concentrated spray solution enhanced the efficacy duration.

As can be seen from the above research, the current research on the application of plant protection UAV is mainly focused on the effect of operation parameters of aerial spraying on droplet deposition distribution. However, a large number of studies have shown that droplet size is the main factor causing droplet deposition and drift among the affecting factors for pesticide spraying [14–18]. The smaller the droplet, the longer it floats in the air and the more effective it is in drifting with the wind [19,20]. Martin et al. [21] carried out a study on the distribution of droplet spraying by aerial electrostatic nozzles with different orifice sizes and droplet size spectrum under different wind speed conditions by using a wind tunnel. Ferguson et al. [22] measured droplet density, droplet coverage, and deposition penetration of different sizes of droplets sprayed by different types of nozzles on the canopy of oat plants through a laboratory spray test. For manned agricultural aircraft and ground plant protection machinery, Hoffmann and Fritz et al. [23–25] tested and evaluated the droplet anti-drift ability of five kind of droplets with different sizes in high-speed wind tunnels and low-speed wind tunnels, respectively. To provide data guidance for an aerial spraying application program of manned agricultural fixed-wing aircraft, Tang et al. [26] studied the atomization characteristics of a normal flat fan nozzle and air induction nozzle under high speed airflow conditions. For obtaining the spray particle size distribution

of GP-81A series nozzles for aerial spraying, Ru et al. [27] carried out the droplet size distribution test under wind tunnel conditions and flight conditions.

At present, the research on the influence of droplet size on the distribution characteristics of droplet deposition were mainly carried out in indoor wind tunnels, and these studies were mainly for ground machinery [28–30] and manned agricultural aircraft [31,32]. Due to the presence of a wind field generated by the rotors of plant protection UAV, the influence of droplet size on droplet deposition and drift would be more significant [33–35], and droplet size needs to be regarded as the most important factor for the research of aerial spraying technology. However, it is interesting to note that there is no report about the effects of droplet size on the distribution of droplet deposition in aerial spraying by using plant protection UAVs under the actual field conditions. For this reason, similarly to that proposed by other authors in orchard spray applications using ground machinery [36], an aerial spraying test with the same spraying rate and different sized droplets in rice canopy was carried out by using multi-rotor UAV (MG-1S, DJI) and four TEEJET nozzles with different orifice sizes in this paper, and the effect of droplet sizes on the distribution of droplet deposition was analyzed by comparing the deposition and drift of droplets of different sizes with the same spray flow rate on rice canopy. It is hoped that the results have significance for optimizing the droplet parameters of aerial spraying, increasing the spraying efficiency, and realizing precision agricultural aviation spray.

## 2. Materials and Methods

### 2.1. Instruments and Equipment

Plant Protection UAV and its Spraying System

As shown in Figure 1, the model of UAV used in this spraying test was MG-1S eight-rotor electric unmanned aerial vehicle (Shenzhen DJI Technology Co., Ltd., ShenZhen, China). The UAV was powered by 12,000 mAh Li-Po batteries. The flying time was about 10–15 min with a full tank, and the capacity of the tank was 10 L. The flight speed was 1–7 m/s and the effective spraying width was 5.0 m. The type of UAV has four hydraulic nozzles, which are symmetrically distributed on both sides of the fuselage. During spraying operation, the latter two nozzles are used for spraying. The test nozzles are Teejet series flat fan nozzles characterized by four different orifice sizes, namely Teejet 11001VS, Teejet 110015VS, Teejet 11002VS, and Teejet 11003VS. The flow rate of the spraying system can be adjusted by a hand-held ground station. The spraying flow rate used in this test was adjusted to 1.4 L/min, that is, the flow rate of a single nozzle was 0.7 L/min.

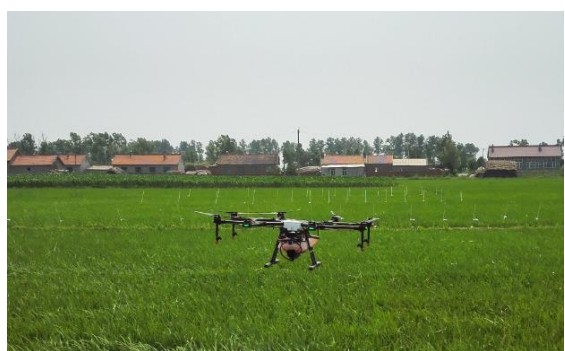

**Figure 1.** Plant protection unmanned aerial vehicle (UAV).

### 2.2. Experiment Design

2.2.1. Field Plots

The tests were conducted at an agricultural experiment station located at Dehui, Jilin Province, China (latitude 126.300223, longitude 44.585429). The tested crop was "Japonica 85" rice in the

tillering stage 29 [37]. Rice was transplanted by machinery with a row spacing between rice plants of 17 cm × 14.5 cm and an average height of 55 cm. The rice in the whole test area grew well and consistently.

### 2.2.2. Sampling Scheme

The test sampling scheme is shown in Figure 2. As three repetitions, droplet collectors arranged in three lines spaced equally apart and the same sampling point arrangement were set up in the test area, and the distance between the collectors lines was 15 m. The collection point directly below the UAV route was recorded as 0 m, and based on the direction of the ambient wind, four collection points equal distances apart were spaced at −1, −2, −3, and −4 m on the upwind side of the flight route, ten collection points were spaced at 1, 2, 3, 4, 5, 7, 9, 11, 13, 15 m on the downwind side of the flight route. Mylar cards were fixed horizontally on plastic rods by using double-head clamps at each collection point, which were used to measure the droplet deposition distribution in the canopy. The height of these Mylar cards was adjusted to the head of the rice canopy by clamps. In addition, in order to analyze the penetration of droplets with different sizes, droplet deposition in the lower layer of rice plants in the effective spray area needed to be collected. Based on the effective spray width of 5 m for the test plant protection UAV, the range of effective spraying area in these tests can be set to −2.5 m to 2.5 m. It is necessary to place Mylar cards to collect droplets in the lower layer of rice plants at these collection points of −2, −1, 0, 1, 2 m. Therefore, the collection point at 3 m in the downwind direction is used as the starting point of drift area, which is recorded as 0 m.

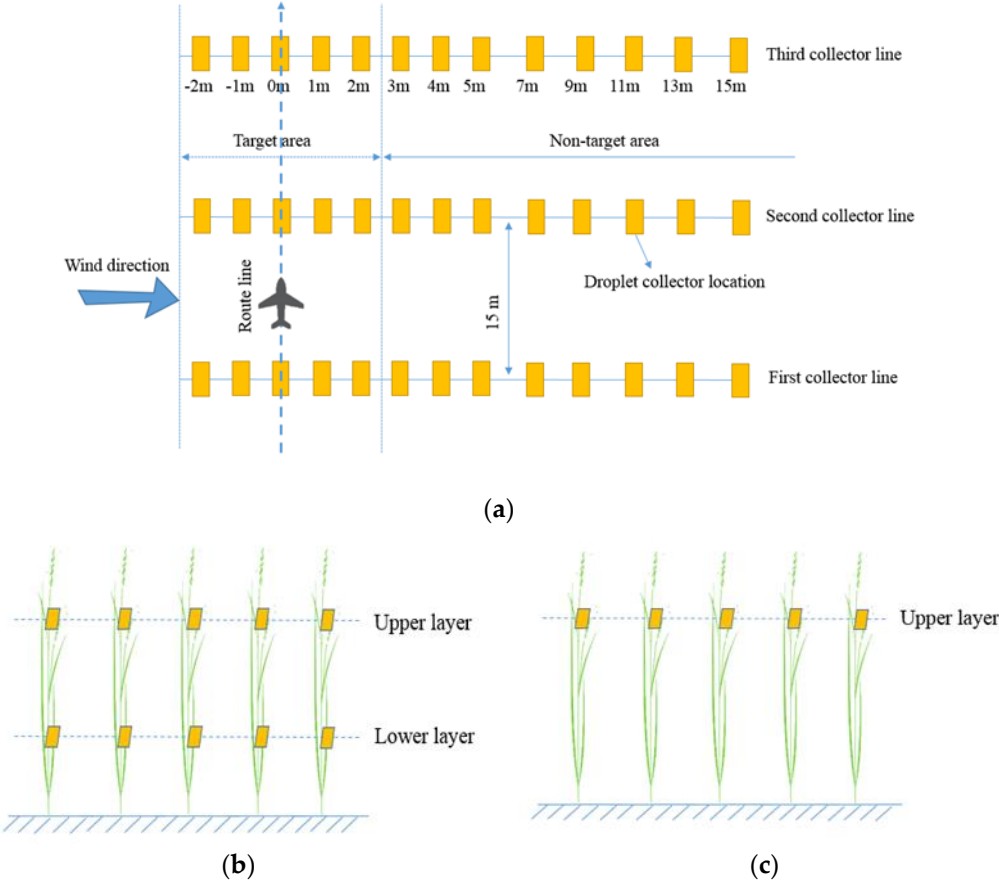

(**a**)

(**b**)　　　　(**c**)

**Figure 2.** (**a**) Overall experimental layout of each treatment; (**b**) Placement of Mylar cards at target area; and (**c**) Placement of Mylar cards at non-target area.

### 2.2.3. Operation Parameters

In all tests, allure red (purchased from Beijing Oriental Care Trading L.,td., Beijing, China) was used as the tracer in spraying solution with a concentration of 10 g/L(only water + tracer). Allure red, a water-soluble colorant, is frequently used in these types of studies. It presents high recovery rate, high photostability, and low acute toxicity in different species of animals according to the Joint FAO/WHO Expert Committee on Food Additives, as well as the European Union's Scientific Committee for Food.

In order to ensure the authenticity and validity of the test results, normal operation parameters (flight speed of 5 m/s, flight height of 1.5 m) and operation mode with fully autonomous plant protection UAV were selected for this spray test. The environmental meteorological parameters were recorded by a Hberw6-3 portable ultrasound Micro meteorological station (Shenzhen hongyuanbo Technology Co., Ltd., Shenzhen, China), which was used to monitor the wind speed and direction, temperature, and humidity of environment during the spraying test. The measurement accuracy of the meteorological station is (±2%), and the recording frequency for weather parameters of the meteorological station is 1 Hz. The test time was 7:00–9:00 a.m. The environmental data collected during the aerial spraying time are presented in Table 1.

**Table 1.** The environmental conditions during the aerial spraying test.

| Test Treatment | Air Temperature/(°C) | Air Humidity/% | Wind Speed/(m/s) |
|:---:|:---:|:---:|:---:|
| T1 | 23.6~24.1 | 55.4~56.3 | 2.0~2.7 |
| T2 | 23.7~24.1 | 55.6~56.0 | 1.9~2.6 |
| T3 | 24.2~24.7 | 56.4~56.9 | 1.8~2.3 |
| T4 | 24.8~25.1 | 56.8~57.6 | 1.9~2.5 |

### 2.3. Droplet Size Measurement

In order to ensure that the droplet deposition results are comparable, it is necessary to change the spray pressure of the pump to make sure the spray flow rate of four nozzles with different orifice sizes are the same during the spray test. The device shown in Figure 3 is a spraying test system built by the author (specify) to measure the flow range of nozzles. The flowmeter used in the system is a FLR1000 low flow turbine flowmeter (OMEGA Engineering Inc., Norwalk, CT, USA) with a measurement accuracy of ±1%, and the pump used in the system is a PLD-1205 diaphragm pump (PULANDI Electrical and Mechanical Equipment Co., Ltd., Hebei, China) with an accuracy of ±3%. Based on the basic principle that impeller rotation speed and flow of pump changed with input voltage, the flow range of four nozzles with different orifice sizes can be measured by continuously changing the input voltage of the pump. According to the results of three repeated tests, the spraying flow ranges of the four nozzles were 0–780, 0–980, 0–1200, and 0–1390 mL/min, respectively.

According to the measured nozzle flow range, 700 mL/min was selected as the flow rate of single nozzle in the test. The spray pressure of nozzle TEEJET 11001VS, TEEJET 110015VS, TEEJET 11002VS, TEEJET 11003VS were 0.50, 0.38, 0.25, and 0.15 MPa, respectively, when the flow rate of each nozzle is 700 mL/min.

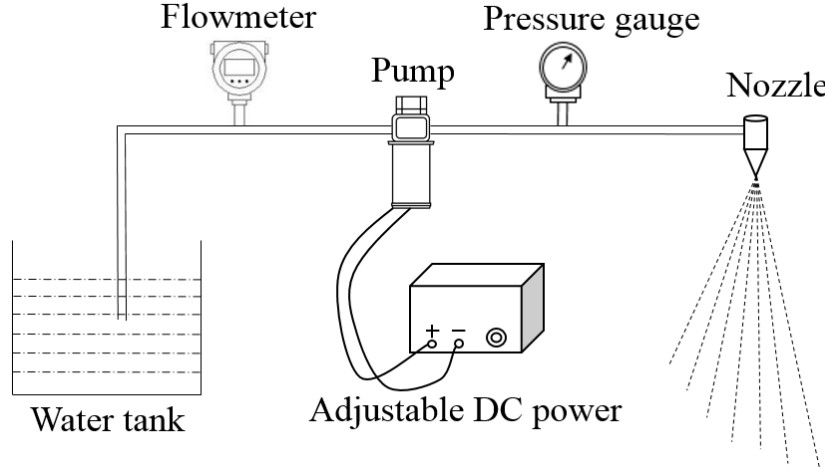

**Figure 3.** Spray test system.

The droplet size sprayed by nozzles with different orifice sizes should be measured with different spray pressures under the same flow of nozzles. The measurement site was shown in Figure 4. The equipment used for the measurement of droplet size is a laser particle size analyzer (DP-2, OMEC Instruments Co., Ltd., Zhuhai, China). The distance between the transmitting end and the receiving end of the laser particle size analyzer is 120 cm, and the distance between the nozzle and the measuring laser beam is 40 cm to ensure that droplets are fully atomized. The measurement results of droplets were classified based on the ASAE (American Society of Agricultural Engineers) S-572 standard, which was shown in Table 2.

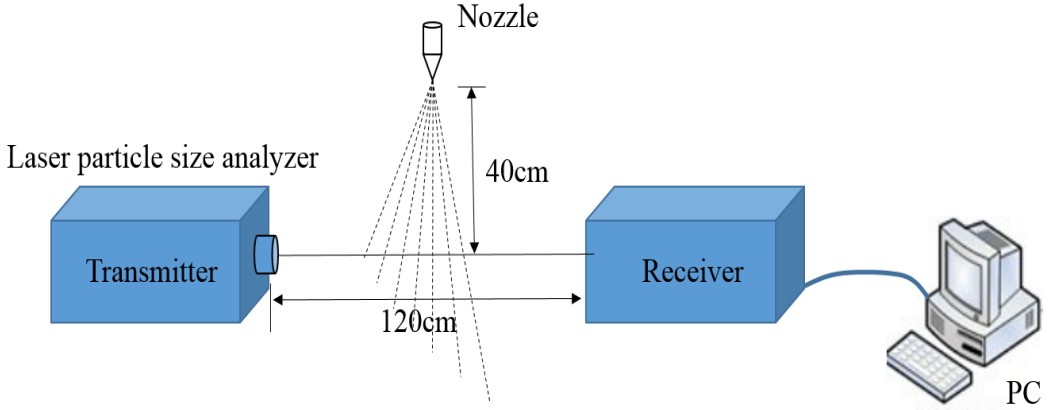

**Figure 4.** Droplet size measurement.

**Table 2.** Results of droplet size measurement and test treatment.

| Nozzle Type | Spray Pressure /MPa | VMD/μm | Dv0.1/μm | Dv0.9/μm | Droplet Classification | Test Treatment |
|---|---|---|---|---|---|---|
| TEEJET 11001VS | 0.50 | 95.21 | 50.85 | 175.16 | Very-Fine | T1 |
| TEEJET 110015VS | 0.38 | 121.43 | 60.45 | 217.45 | Fine | T2 |
| TEEJET 11002VS | 0.25 | 147.28 | 69.47 | 235.80 | Fine | T3 |
| TEEJET 11003VS | 0.15 | 185.09 | 87.53 | 365.73 | Medium | T4 |

*2.4. Sample Processing and Statistical Analysis*

Nearly 30 s after spraying, all Mylar cards were removed and placed in self-sealing bags along with a label describing the treatment, replication, and Mylar cards site information. For each plot, 54 samples were placed into light-proof sealed boxes immediately after collection and transported

to the laboratory for analysis. Each Mylar card sample was washed in 20 mL of distilled water in the collection bags, respectively. Samples were agitated and vibrated for 10 min to allow the dye to dissolve into the water solution. Previous tests have shown that this methodology results in near total recovery of dye deposited on samples [38]. After vibration and elution, the solution was poured into a cuvette to measure the absorbance value by a UV755B ultraviolet-visible spectrophotometer (Shanghai Yoke Instrument, Co. Ltd., Shanghai, China) at an absorption wavelength of 514 nm. Spray deposits were quantified by comparison with similarly determined dye concentrations from spray tank samples and the area of the respective samples. The data were expressed as a quantity of dye (μg) deposited per unit area of the sample (cm$^2$) [3].

In order to ensure the measurement accuracy, six concentrations of temptation red standard solutions were set up for calibration and measurement within the absorbance range of ultraviolet-visible spectrophotometer, and linear regression fitting was performed on the relationship between the concentration of temptation red solution and the measurement results of absorbance. By fitting, the determination coefficient R$^2$ was 0.999, which indicates that the concentration of temptation red has a good linear relationship with absorbance in the range of determination.

After all the sample concentration values were measured, the deposition amount and deposition rate of droplets at each sampling point were calculated according to ISO22866 standard [39], and the calculation formula is shown in equation (1):

$$\beta_{dep\%} = 100 \times \frac{\beta_{dep}}{\beta_v} \times 100\% \tag{1}$$

and $\beta_v$ is defined by

$$\beta_v = \frac{Q}{100 \cdot w \cdot s} \tag{2}$$

where $\beta_{dep}$ is the deposition amount of droplet per unit area (μg/cm$^2$), $\beta_{dep\%}$ is the deposition rate of droplet (%), $\beta_v$ is the amount of liquid applied per hectare (g/ha), Q is the spraying flow rate of plant protection UAV (μg/s), w is the effective spray width of plant protection UAV (m), and s is the flight speed of plant protection UAV (m/s).

In order to characterize the droplet deposition penetration at each collection point in the target area, the ratio of the lower droplet deposition rate to the sum of the upper and lower droplet deposition rates at the collection point is used as a measurement for droplet deposition penetration, and the calculation formula is shown in equation (3):

$$P = \beta_{dep2}/(\beta_{dep1} + \beta_{dep2}) \times 100\% \tag{3}$$

where $\beta_{dep1}$ and $\beta_{dep2}$ represent the deposition amount of droplet per unit area (μg/cm$^2$) in the upper and lower layers of each collection point, respectively.

According to the ISO 22866 standard [13,35], the formula for calculating the cumulative drift rate of droplets ($\beta_{T\%}$) is:

$$\beta_{T\%} = \int_0^x \beta_{dep1\%}(x)dx \tag{4}$$

where $\beta_{T\%}$ is the cumulative drift rate of droplets (%), $\beta_{dep1\%}$ represent the droplet deposition rate in upper layer of each collection point (%), and x is the drift distance of droplets (m).

To further demonstrate the effect of droplet size parameters on the droplet deposition and drift of aerial spray in rice canopy, the significant difference for the results of droplet deposition, droplet penetration, and drift were conducted using analysis of variance (One-Way ANOVA) by Duncan's test at a significance level of 95% with SPSS v22.0 (SPSS Inc, an IBM Company, Chicago, IL, USA). More precisely, data are expressed as the mean ± standard deviation (SD) [40].

## 3. Results and Discussion

In the following, the impacts of different droplet sizes on the droplet deposition and droplet penetrability in the target area and droplet drift in non-target area are discussed based on the collected droplet deposition data.

### 3.1. Droplet Deposition Results in Target Area

To obtain a satisfactory spraying efficiency, it is meaningful to optimize the droplet size parameters to obtain better droplet deposition distribution. As shown in Figure 5, due to the influence of external environmental wind, the droplet deposition rate in the target area gradually increases from upwind to downwind, and all of them drift to a certain extent. According to the results of droplet deposition, it can be seen that the peak value of droplets deposition rate of T3 (droplets with a volume median diameter [VMD] of 147.28 µm) and T4 (droplets with a VMD of 185.09 µm) were mainly at the collection location of 1~2 m in the target area, while that of T1 (droplets with a VMD of 95.21 µm) and T2 (droplets with a VMD of 121.43 µm) drift to the non-target area along the environmental wind direction.

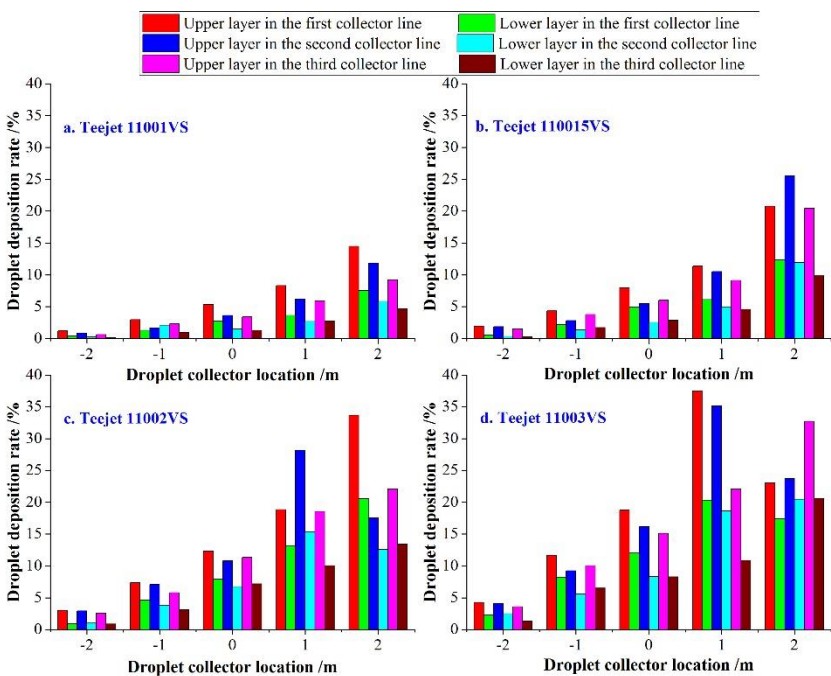

**Figure 5.** Droplet deposition distribution in target area.

The droplet deposition distributions were compared and analyzed (Figure 6). The average droplet deposition rates of T1, T2, T3, and T4 in the upper and lower rice canopy were 5.19%, 8.89%, 13.48%, and 17.81% and 2.48%, 4.43%, 8.09%, 10.89%, respectively. According to the analysis of variance of test data, there are significant differences in the droplet deposition results among T1, T2, T3, and T4. The average deposition rate of T4 in upper and lower rice canopy were the largest, and were significantly higher than that of T3, T2, and T1. It indicated that with increased droplet size, the droplet deposition rate in the upper and lower rice canopy were increased in the target area. Similar to other studies [16,41], the reason for the droplet deposition results difference could be that the droplets with smaller size are more likely to drift outside the target area under the influence of the environmental wind, and the droplets with larger size which are less affected by the environmental wind due to the mass of the droplet are more likely to deposit in the target area.

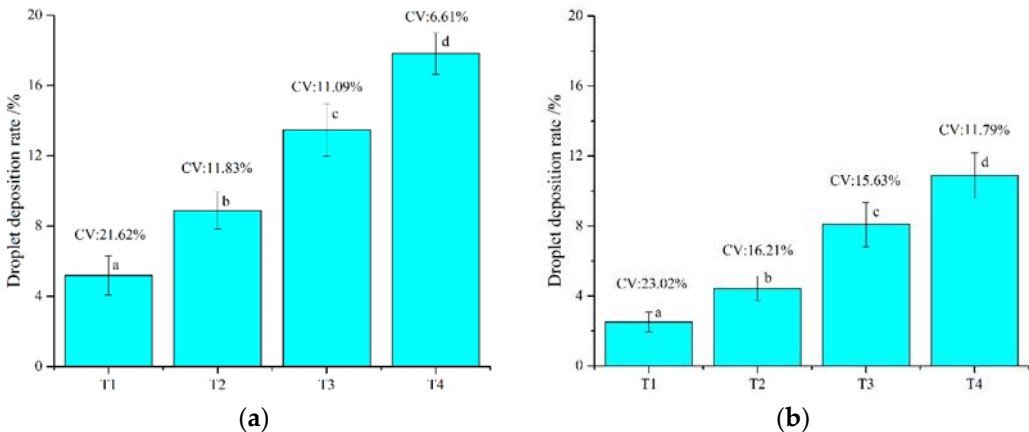

**Figure 6.** (**a**) Average droplet deposition rate in upper layer; and (**b**) Average droplet deposition rate in lower layer. Bars with different letters are significantly different, Duncan's test, *p*-level < 0.05.

A smaller CV value indicates that the distribution of droplets in the rice canopy is more uniform. It can also be seen from Figure 6 that the droplet deposition uniformity of T1, T2, T3, and T4 in the upper rice canopy were 21.62%, 11.83%, 11.09%, and 6.61% respectively, and that of T1, T2, T3, and T4 in lower rice canopy were 23.02%, 16.21%, 15.63%, and 11.79%, respectively. The test of T4 has a better uniformity for droplet deposition. The droplet deposition uniformity increased with the increase of droplet size, which indicated that the increase of droplet size helped to improve the droplet deposition uniformity. It could be that the smaller droplets are more susceptible to the external environmental wind field and rotor turbulence wind field, which makes their deposition irregular and difficult to deposit effectively in the target area.

*3.2. Droplet Deposition Penetration*

To improve the deposition rate of solution and control efficacy, it is crucial to enhance the droplet deposition penetration and obtain a homogeneous deposition distribution, especially since many diseases and pests occur on the bottom of plants [42]. Figure 7 showed the penetration results of droplets with four different size parameters (T1, T2, T3, and T4) in rice canopy. In each test, the penetration results of droplets at each collection location in the target area were almost the same, except for the collection location at -2 m. As can be seen from Figure 5, it was mainly due to the fact that most of the droplets drift to the right side of the flight route under the influence of the environmental wind, and very few droplets deposited at the edge of the left side in the target area, especially in the lower rice canopy at the edge. In addition, the variability of droplet deposition penetration in the four tests (T1, T2, T3, and T4) were 8.36%, 6.79%, 3.29%, and 4.01%, respectively, which were less than 10%. It indicated that the penetration results of droplet deposition in each test were consistent and effective, and tend to be more stable with the increase of droplet size.

According to the analysis results, the average droplet deposition penetration of T1, T2, T3, and T4 in rice canopy were 32.01%, 33.48%, 37.82%, 38.13%, respectively (Figure 8). There were no significant differences in the droplet deposition penetration between T1 (32.01%) and T2 (33.48%), T3 (37.82%) and T4 (38.13%), while the average droplet depositions of T3 (37.82%) and T4 (38.13%) were significantly higher than those of T1 (32.01%) and T2 (33.48%). The droplet deposition penetration also increased with the increase of droplet size, which indicated that the increase of droplet size helped to improve the droplet deposition penetration. However, Ferguson et al. and Wolf et al. [25,43] have tested the deposition penetration of droplets with different sizes by using the ground spraying equipment and showed that droplets with smaller size have better penetration in crop canopy, which was different from our test results. The reason for the droplet deposition results difference could be caused by the difference in the spraying equipment. When the ground machinery is spraying, the droplets with larger size will fall faster under the action of their own gravity, and can reach the crop canopy faster.

But it cannot be ignored that these larger droplets are also easily captured by the upper canopy of the crop, and mainly deposited on the top layer of crop. For the droplets with smaller size, their falling speed is continuously reduced due to their light weight and the air resistance [44,45]. Most of the smaller droplets are free in the air, and more easily reach the lower canopy of the crop. Therefore, smaller droplets may have better penetration than larger droplets for the ground spraying equipment. However, the downwash rotor wind field of plant protection UAV can accelerate the deposition speed of droplets and blow the upper leaves of the crop at the same time, so that a large number of larger droplets with a faster falling speed can reach the lower canopy of the crop within the action time of the rotor wind field [22,23,46]. Therefore, the penetration ability of larger droplets may be better than that of smaller droplets within a certain range of droplet size for the plant protection UAV.

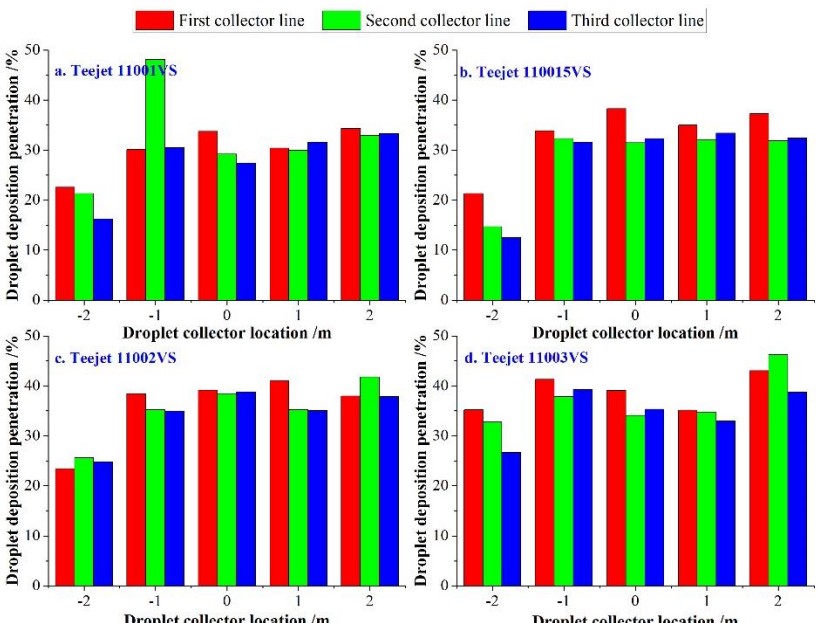

**Figure 7.** Droplet deposition penetration in target area.

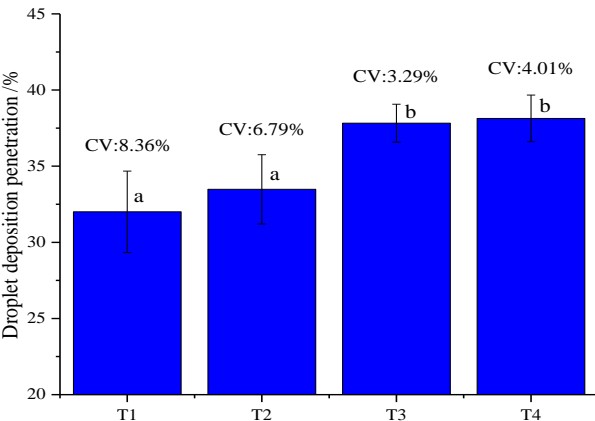

**Figure 8.** Average droplet deposition penetration in target area. Bars with different letters are significantly different, Duncan's test, *p*-level < 0.05.

*3.3. Droplet Drift*

The distribution of drift rate of droplets with different particle sizes at different drift distances was shown in Figure 9. In the test of T1, the peak values of droplet drift rate on the second collection belt and the third collection belt appear at 1.5 m and 2.5 m in the downwind drift area, respectively. The ranges of drift distance of droplets with different sizes in the rice canopy were more than 12 m,

10~12 m, 10~12 m, and 8~10 m, respectively. The drift distance of droplets in the T1 test was the farthest, and the drift distance of droplets gradually decreases as the droplets size increases. Combined with the droplet deposition results in the target area, it is known that the smaller droplets have a greater drift with the environmental wind direction due to the small droplet size and the presence of the environmental wind field.

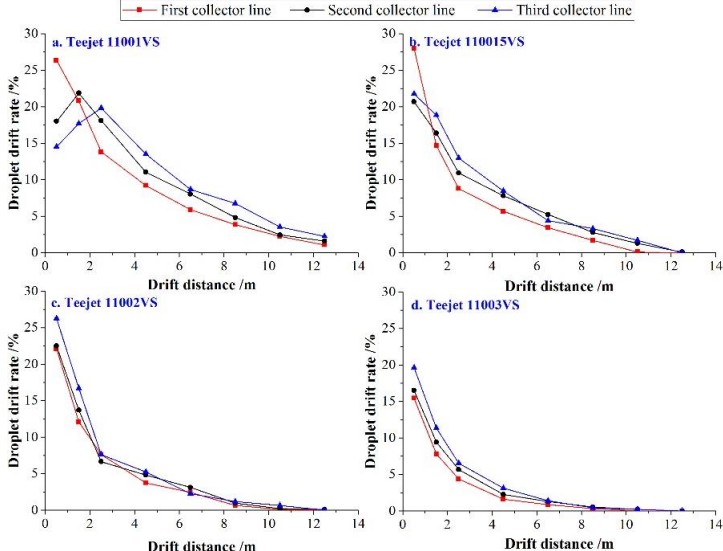

**Figure 9.** Droplet drift in the non-target area.

The cumulative drift rate of droplets refers to the total drift ratio of droplets during spraying, which is used to measure the anti-drift potential of droplets. The smaller the value of the cumulative drift rate, the better the anti-drift ability of droplets. From the calculation of formula (4), the average value of the cumulative drift rate of droplets in rice canopy in the four tests (T1, T2, T3, and T4) were 73.87%, 50.26%, 35.91%, and 23.06%, respectively (Figure 10). According to the analysis results, there were significant differences in the cumulative drift rate of droplets between the four tests, and the cumulative drift rate of droplets of T1 (73.87%) was significantly higher than that of T2 (50.26%), T3 (35.91%), and T4 (23.06%). The cumulative drift rate of droplets decreased sharply with the increase of droplet size, which indicated that droplet size have an extremely significant effect on droplet drift in the rice canopy and the increase of droplet size can effectively reduce droplet drift.

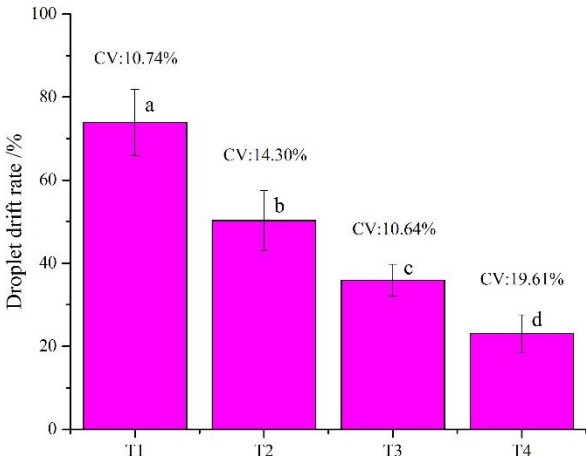

**Figure 10.** Cumulative drift rate of droplet. Bars with different letters are significantly different, Duncan's test, *p*-level < 0.05.

To further analyze the relationship between the result of droplet drift and droplet size, correlation analysis and regression analysis were conducted between the cumulative drift rate and the VMD of droplets (Figure 11). The results showed that there was a significant negative correlation between the cumulative drift rate of droplets and the VMD of droplets within a certain range of droplet size ($p < 0.05$). Furthermore, according to the standard for the quality of agricultural aviation operation (MHT 1002-1995) [47], the cumulative drift rate of droplets in agricultural aviation spraying operation should be less than 30%. Therefore, for the application of plant protection UAV with extra-low volume spraying, in order to reduce droplet drift and improve the effect of droplet deposition, the use of droplets with Dv0.5 less than 160 μm should be avoided and a more than 10 m buffer zone should be considered downwind of the spraying field to avoid the drug damage caused by pesticide drift.

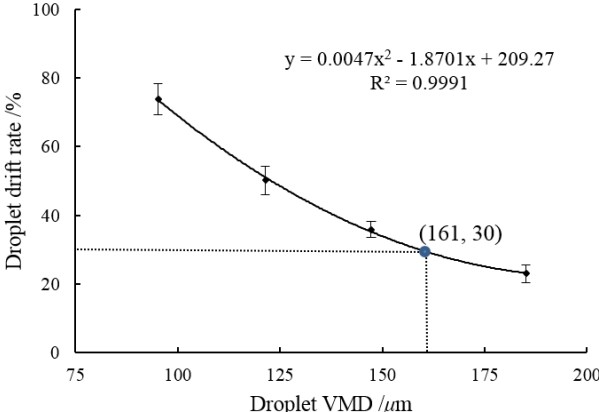

**Figure 11.** Regression analysis between the cumulative drift rate and the volume median diameter (VMD) of droplets.

## 4. Conclusions

In this study, four aerial spraying tests with the same flow rate and differently sized droplets in a paddy field was carried out by using a multi-rotor plant protection UAV and four TEEJET nozzles with different orifice sizes. The deposition distribution and penetration of droplets in the target area and the drift distribution of droplets in the non-target area were compared and analyzed in this research. The conclusions are shown as follows:

(1) The deposition distribution and penetration of droplets in the target area and the drift distribution of droplets in the non-target area were influenced by the droplet size.

(2) There were significant differences in the droplet deposition rate in the target area between the four tests. The average deposition rate of droplets with a VMD of 185.09 μm in the upper and lower rice canopy were the largest, and the droplet deposition rate in the upper and lower rice canopy were increased in the target area with the increase of droplet size.

(3) The penetration results of droplets also increased with the increase of droplet size, and that of droplets with a VMD of 185.09 μm was the best, reaching 38.13%. The penetration results of droplets sprayed by plant protection UAV are different from those of the ground spraying machinery.

(4) The average value of the cumulative drift rate of droplets in the rice canopy in the four tests were 73.87%, 50.26%, 35.91%, and 23.06%, respectively, and the cumulative drift rate and the drift distance of droplets decreased with the increase of droplet size, which indicated that the increase of droplet size can effectively reduce droplet drift.

The experiment demonstrated that the droplet size is one of the most important factors affecting droplet deposition and drift for pesticide spraying by plant protection UAV. For the application of plant protection UAV with extra-low volume spraying, in order to reduce droplet drift and improve the effect of droplet deposition, the use of droplets with VMD less than 160 μm should be avoided and a more than 10 m buffer zone should be considered downwind of the spraying field to avoid the drug

damage caused by pesticide drift. In addition, due to the existence of the rotor wind field of plant protection UAV, the deposition movement of droplets sprayed by plant protection UAV is different from that of the ground spraying machinery. Therefore, the effect of the rotor flow field of UAV on the deposition movement and distribution characteristics of droplets with different size should be studied to achieve precision agricultural aerial spraying in the future.

**Author Contributions:** S.C., Y.L. and Z.Z. conceived the idea of the experiment. S.C. and X.H. built the spraying test system, performed the indoor experiments and analyzed the data. S.C., G.W. and S.C. performed the field test and analyzed the data. S.C., X.D. and F.O. wrote and revised the paper. All authors have read and agreed to the published version of the manuscript.

**Funding:** The study was funded by the science and technology planning project of Guangdong Province (2017B010116003, 2019B020208007, 2017B010117010), the National Natural Science Foundation of China (Grant No.61773171), the young innovative talents project of regular institutions of higher education of Guangdong Province (2018KQNCX020), the leading talents program of Guangdong Province (2016LJ06G689).

**Conflicts of Interest:** The authors declare no conflict of interest.

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
