# Peer review of "Effect of Droplet Size Parameters on Droplet Deposition and Drift of Aerial Spraying by Using Plant Protection UAV"

_agronomy, doi:10.3390/agronomy10020195_

Round 1
Reviewer 1 Report
GENERAL COMMENTS TO THE MANUSCRIPT:
In general, the use of UAV for aerial spray application with relative resulting canopy deposition is a research hot-topic in precision agriculture all around the world and especially in China where UAV spray application is becoming more and more preferred. The paper provide a very comprehensive overview of spray application process in arable crops using UAV spray application technique. The contents of manuscript are relevant and important information about the effect of droplet size spectra on both canopy spray deposition and spray drift could be obtained.
In general the paper is well organized and comprehensive without any compromising lack. However, considering the accurate drift measurements performed, I strongly suggests to add a table in which the environmental parameters recorded are reported separately for each tests; the specific weather information will increase the quality of manuscript adding a further fundamental data about drift process generation during spray application.
From my point of view, I recommend accept and publishing the manuscript. I have detailed few minor changes required before the publication.
L98: change to read “for ground machinery [28-31] and manned agricultural aircraft[32,33]. Due to the…”
Add the suggested references to support your sentence using the reference formatting required by the journal. The three suggested references will be the numbers 28, 29, 30, 31 and 32; change the references numbering throughout the text:
Balsari, P., Grella, M., Marucco, P., Matta, F., Miranda-Fuentes, A., 2019. Assessing the influence of air speed and liquid flow rate on the droplet size and homogeneity in pneumatic spraying. Pest Management Science 75, 366-379. Nuyttens, D., Baetens, K., De Schampheleire, M., Sonck, B., 2007. Effect of nozzle type, size and pressure on spray droplet characteristics. Biosystems Engineering 97(3), 333-345. Zande, J.C. van de, Holterman, H.J., Wenneker, M., 2008. Nozzle classification for drift reduction in orchard spraying: identification of drift reduction class threshold nozzles. Agricultural Engineering International: the CIGR Ejournal. Manuscript ALNARP 08 0013. Vol. X. May, 2008. Martin, D.E., Carlton, J.B., 2012. Airspeed and orifice size affect spray droplet spectra from an aerial electrostatic nozzle for rotary wing applications. Atomization and Sprays 22(12), 997-1010. Fritz, B.K., Hoffmann, W.C., Kruger, G.R., Henry, R.S., Hewitt, A., Czaczyk, Z., 2014. Comparison of drop size data from ground and aerial application nozzles at three testing laboratories. Atomization and Sprays 24(2), 181-192.
L103: “For this reason, aerial spraying…” change to read “For this reason, similarly to that proposed by other authors in orchard spray application using ground machinery [33], aerial spraying…”. Add the suggested reference using the reference formatting required by the journal. The reference will be the number 36; change the references numbering throughout the text:
Cross, J.V., Walklate, P.J., Murray, R.A., Richardson, G.M., 2001. Spray deposits and losses in different sized apple trees from an axial fan orchard sprayer:2, Effects of spray quality. Crop Protection 20, 333-343.
L119: “…series fan nozzles with four kinds of aperture,…” change to read “…series flat fan nozzles characterized by four different orifice sizes,…”.
L129: “…tillering stage.” Please specify the growth stage using the BBCH scale and add also the reference. I put below the reference of BBCH scale to be used and listed among other references using the formatting requested by the journal (the reference will be the number 37):
Meier, U. Growth Stages of Mono- and Dicotylodonous Plants: BBCH Monograph, 2nd ed.; Uwe Meier Federal Biological Research Centre for Agriculture and Forestry: Braunschweigh, Germany, 2001; pp. 1–158.
L133: “…, three droplet collection belts with…” change to read “…, droplet collectors arranged in three lines with…”.
L135: “…collection belts was…” change to read “…collectors lines was…”. Use “collector line” instead of “collection belt” also in the Figure 2.
L153: “…was 10m. The…” in the Figure 2 the distance draw is 15m. Please make consistent the info between figure and text.
L165-168: In this type of trials where you have a strong influence of environmental conditions, especially wind speed and direction, you cannot rely on general mean data. Please put in a table all the weather data recorder during field trials split for each test performed. According to the ISO22866:2005 you need a weather station able to record weather parameter at 1Hz frequency (1 record per second) in order to describe the weather parameters at the time of field trials in a suitable way. Please add also the frequency of record of your weather parameters between lines 165-166 where the accuracy is detailed.
L187: “…with different apertures should…” change to read “…with different orifice size should…”. Please make this change consistent throughout the text using always the word orifice instead of aperture.
L198: In Table 1 add also the values of D10, D90 and V100 as well recognized the main characteristics and fundamental parameters to describe nozzles at scientific level.
L204: “…in 0.02 L of…” change to read “…in 20 ml of…”.
Author Response
Dear reviewer,
Thank you for your review and insightful comments on our paper. The responds to your comments are as follow. According to your suggestion, the paper has been revised accordingly. Revised version and a copy that highlights the changes to the text in red were attached in the attached file. Please let us know if you have any other ideas or suggestions.
Looking forward to hear from you!
Thanks for your support!
In general the paper is well organized and comprehensive without any compromising lack. However, considering the accurate drift measurements performed, I strongly suggests to add a table in which the environmental parameters recorded are reported separately for each tests; the specific weather information will increase the quality of manuscript adding a further fundamental data about drift process generation during spray application.
Response : Revised. ‘Table 1. The environmental conditions during the aerial spraying test’ has been added to my revised manuscript.
L98: change to read “for ground machinery [28-31] and manned agricultural aircraft[32,33]. Due to the…”
Add the suggested references to support your sentence using the reference formatting required by the journal. The three suggested references will be the numbers 28, 29, 30, 31 and 32; change the references numbering throughout the text:
Balsari, P., Grella, M., Marucco, P., Matta, F., Miranda-Fuentes, A., 2019. Assessing the influence of air speed and liquid flow rate on the droplet size and homogeneity in pneumatic spraying. Pest Management Science 75, 366-379.
Nuyttens, D., Baetens, K., De Schampheleire, M., Sonck, B., 2007. Effect of nozzle type, size and pressure on spray droplet characteristics. Biosystems Engineering 97(3), 333-345.
Zande, J.C. van de, Holterman, H.J., Wenneker, M., 2008. Nozzle classification for drift reduction in orchard spraying: identification of drift reduction class threshold nozzles. Agricultural Engineering International: the CIGR Ejournal. Manuscript ALNARP 08 0013. Vol. X. May, 2008.
Martin, D.E., Carlton, J.B., 2012. Airspeed and orifice size affect spray droplet spectra from an aerial electrostatic nozzle for rotary wing applications. Atomization and Sprays 22(12), 997-1010.
Fritz, B.K., Hoffmann, W.C., Kruger, G.R., Henry, R.S., Hewitt, A., Czaczyk, Z., 2014. Comparison of drop size data from ground and aerial application nozzles at three testing laboratories. Atomization and Sprays 24(2), 181-192.
Response : Revised. The suggested references has been added to my revised manuscript.
L103: “For this reason, aerial spraying…” change to read “For this reason, similarly to that proposed by other authors in orchard spray application using ground machinery [33], aerial spraying…”. Add the suggested reference using the reference formatting required by the journal. The reference will be the number 36; change the references numbering throughout the text:
Cross, J.V., Walklate, P.J., Murray, R.A., Richardson, G.M., 2001. Spray deposits and losses in different sized apple trees from an axial fan orchard sprayer:2, Effects of spray quality. Crop Protection 20, 333-343.
Response : Revised
L119: “…series fan nozzles with four kinds of aperture,…” change to read “…series flat fan nozzles characterized by four different orifice sizes,…”.
Response : Revised
L129: “…tillering stage.” Please specify the growth stage using the BBCH scale and add also the reference. I put below the reference of BBCH scale to be used and listed among other references using the formatting requested by the journal (the reference will be the number 37):
Meier, U. Growth Stages of Mono- and Dicotylodonous Plants: BBCH Monograph, 2nd ed.; Uwe Meier Federal Biological Research Centre for Agriculture and Forestry: Braunschweigh, Germany, 2001; pp. 1–158.
Response : Revised
L133: “…, three droplet collection belts with…” change to read “…, droplet collectors arranged in three lines with…”.
Response : Revised
L135: “…collection belts was…” change to read “…collectors lines was…”. Use “collector line” instead of “collection belt” also in the Figure 2.
Response : Revised
L153: “…was 10m. The…” in the Figure 2 the distance draw is 15m. Please make consistent the info between figure and text.
Response : Revised
L165-168: In this type of trials where you have a strong influence of environmental conditions, especially wind speed and direction, you cannot rely on general mean data. Please put in a table all the weather data recorder during field trials split for each test performed. According to the ISO22866:2005 you need a weather station able to record weather parameter at 1Hz frequency (1 record per second) in order to describe the weather parameters at the time of field trials in a suitable way. Please add also the frequency of record of your weather parameters between lines 165-166 where the accuracy is detailed.
Response : Revised
L187: “…with different apertures should…” change to read “…with different orifice size should…”. Please make this change consistent throughout the text using always the word orifice instead of aperture.
Response : Revised
L198: In Table 1 add also the values of D10, D90 and V100 as well recognized the main characteristics and fundamental parameters to describe nozzles at scientific level.
Response : Revised
L204: “…in 0.02 L of…” change to read “…in 20 ml of…”.
Response : Revised
Reviewer 2 Report
See the attached file.

Author Response
Dear reviewer,
Thank you for your review and insightful comments on our paper. The responds to your comments are as follow. According to your suggestion, the paper has been revised accordingly. Revised version and a copy that highlights the changes to the text in red were attached in the attached file. Please let us know if you have any other ideas or suggestions.
Looking forward to hear from you!
Thanks for your support!
Respond
General observation
The Authors present a paper entitled “Effect of Droplet Size Parameters on Droplet Deposition and Drift of Aerial Spraying by Using Plant Protection UAV”. The study is very interesting, because it provides useful information to optimise phytosanitary treatments and reduce the negative effect of plant protection products on the environment in the light of precision agriculture principles. I think the manuscript, after a proper revision, is worth to be published. In particular, some aspects of materials and methods should be better described and the statistical analysis of deposits should be carried out more in depth.
General comments.
Key words should be not already included in the title.
Response : Revised
Some references (3, 6,16, 20) are written with main words in capital letters, others in lower case. Beconsistent with journal reference style.
Response : Revised
Although I am not in a position to judge the English language, a revision is recommended.
Response : Modified
Specific comments.
Line 29: … diameter (VMD) …
Response : Revised
Lines 30-31: cumulative drift rates should be referred to the nozzle’s features, so I suggest you add VMD values at Line 23.
Response : Revised
Lines 58-63: Authors should add some comments by observing that in other Countries (European Union) “aerial spraying of pesticides has the potential to cause significant adverse impacts on human health and the environment, in particular from spray drift. Therefore, aerial spraying should generally be prohibited …” (EU Directive 2009/128/EC).
Response : This part is mainly used to describe the advantages of plant protection UAV. I think it is maybe not suitable to put the drift caused by aerial spraying here. In addition, few other countries in Europe use plant protection UAV to spray pesticides.
Line 75: … (such as droplets of …
Response : Revised
Line 80: … is mainly focused …
Response : Revised
Line 87: … droplets sprayed by …
Response : Revised
Lines 113-122: adjust line spacing.
Response : Revised
Line 115: … powered by 12,000 mAh Li-Po batteries.
Response : Revised
Line 117: … The type of UAV has four …
Response : Revised
Lines 119-120 and subsequent: …Teejet 11015VS… I think it should be Teejet 110015VS.
Response : Revised
Line 122: … was 0.7 L/min. (leave a space between value and unit).
Response : Revised
Line 125: Experiment design
Response : Revised
Line 126: Field plots
Response : Revised
Line 128: The tested crop was …
Response : Revised
Line 129: … tillering stage. Report BBCH code.
Response : Revised
Lines 133-153: Authors should specify that spraying tests were carried out considering a single pass of the UAV and that during each pass one nozzle type was selected. As a consequence, the three collection belts can’t be considered replicates (from the statistical point of view) because they aren’t randomised and deposits are correlated. Finally, Authors should add some comments on the difference between experimental tests (single pass) and real treatments (multiple passes, with overlap of deposits due to drift).
Response : Because the apart of each collectors lines was 15m, and the collectors lines are all independent samples, Some researchers[1,2] think that the three collectors lines can be considered replicates. And I would improve my follow-up research based on your comments. Thank you for your helpful comments.
[1] Yao Weixiang, Lan Yubin, Wang Juan, et al. Droplet drift characteristics of aerial spraying of AS350B3e helicopter[J]. Transactions of the Chinese Society of Agricultural Engineering (Transactions of the CSAE), 2017, 33(22): 75-83.
[2]Chen Shengde, Lan Yubin, Li Jiyu, et al. Effect of spray parameters of small unmanned helicopter on distribution regularity of droplet deposition in hybrid rice canopy[J]. Transactions of the Chinese Society of Agricultural Engineering (Transactions of the CSAE), 2016, 32(17): 40-46.
Line 135: … collection belt was 10 m. Moreover, according to the Fig. 2 (a), the distance should be 15 m. Please, check the correctness.
Response : Revised
Line 140: The height of these Mylar cards was adjusted …
Response : Revised
Line 144: … set to -2.5 m to ….
Response : Revised
Lines 155-156: Authors should specify whether the sprayed mixture was only water + tracer. In real treatments, the mixture contains some adjuvants, that heavily affect drop pulverisation.
Response : Revised
Line 155: … (purchased from …
Response : Revised
Line 161: … of 1.5 m) …
Response : Revised
Line 162: … parameters were …
Response : Revised
Line 167: … The test time was …
Response : Revised
Line 172: … built by the author (specify)
Response : Revised
Line 177: … impeller rotation speed …
Response : Revised
Lines 180-181: use always the same symbol for litre (L) (not l).
Response : Revised
Line 184: … was 700 mL/min.
Response : Revised
Lines 191-192: … 200 cm … 50 cm… According to Fig. 4, the distances should be 120 cm and 40 cm, respectively. Please, check the correctness.
Response : Revised
Line 194: … which was shown in Table 1.
Response : Revised
Table 1: Complete Table 1 by adding measurement pressure, Dv0.1 and Dv0.9. This because drift depends not only on the Dv0.5 (VMD), but on the entire droplet spectrum.
Response : Revised
Line 204: … Each Mylar card sample was washed …
Response : Revised
Line 214: … solutions were set up …
Response : Revised
Line 217: … was 0.999 …
Response : Revised
Line 222: according to the units given in the manuscript, equation (1) should be corrected as:
Response : Revised
Line 224: according to the units given in the manuscript, equation (2) should be corrected as:
Response : Revised
Lines 233-234: even if the results are the same, I suggest you define deposition penetration as:
Response : Revised
Lines 240-245: See my previous comments on Lines 133-153 about replicates. Moreover, by using the one-way ANOVA, effects of droplet collection location and layer are not considered. A more general approach could be a three-way ANOVA, considering as sources of variation nozzle, location and layer. Moreover, some comments on the prerequisites of the ANOVA should be reported in the manuscript (normality of the residues, homogeneity of variances). Finally, Authors should specify the mean separation test they applied (protected LSD or Duncan) (not both).
Response : Revised partly.
Lines 251-252: This is a general comments, not a result.
Response : Revised
Lines 259-260: I suggest you to modify the sentence as: “The droplet deposition distributions were compared and analyzed (Fig.6)”.
Response : Revised
Line 262: … test data …
Response : Revised
Fig. 5b: Check nozzle 110015VS.
Response : Revised
Line 285: Add the adopted mean separation criterion: “mean separation at p-level = 0.05 by LSD or Duncan test”.
Response : Revised
Lines 305-306: … have tested …
Response : Revised
Fig. 7b: Check nozzle 110015VS.
Response : Revised
Line 325: see my previous comments on Line 285.
Response : Revised
Fig 9: Correct all nozzle codes.
Response : Revised
Fig. 10: see my previous comments on Line 285. Moreover, correct letters on the bars (all treatments are statistically different).
Response : Revised
Lines 358 and 385: use always VMD (rather than Dv0.5).
Response : Revised
Round 2
Reviewer 2 Report
There are some minor corrections only to be carried out:
Line 203: Authors should specify which author built the spraying test system.
Lines 350, 398, and 422: ... are significantly different ...
Author Response
Dear reviewer,
Thank you for your review and insightful comments on our paper again. The responds to your comments are as follow. According to your suggestion, the paper has been revised accordingly. Revised version was attached in the attached file. Please let us know if you have any other ideas or suggestions.
Looking forward to hear from you!
Thanks for your support! Happy Chinese new year!
Best,
Shengde Chen
Respond
Line 203: Authors should specify which author built the spraying test system.
Response : Revised
Lines 350, 398, and 422: ... are significantly different ...
Response : Revised